# Prevalence of intestinal parasitic infections and associated risk factors among patients attending Debarq Primary Hospital, northwest Ethiopia

Amir Alelign[1]*, Nigus Mulualem[1], Zinaye Tekeste[2]

1 Department of Biology, College of Natural and Computational Science, University of Gondar, Gondar, Ethiopia, 2 Aklilu Lemma Institute of Pathobiology, Addis Ababa University, Addis Ababa, Ethiopia

* aleamiro2009@gmail.com

**Data Availability Statement:** The data supporting the findings of this study are available within the manuscript.

## Abstract

Intestinal parasitic infections (IPIs) are major public health problems in developing countries. This study was conducted to determine the prevalence and associated risk factors of IPIs at Debarq Primary Hospital in northwest Ethiopia. A health facility-based cross-sectional study was conducted from March 2022 to June 2022. The study participants were recruited from patients who visited Debarq Primary Hospital during the study period. Stool samples were collected from each participant and examined for intestinal parasites using direct wet-mount and formal-ether concentration techniques. Socio-demographic data were collected using a structured questionnaire. Out of 422 individuals examined, 33.64% were infected with at least one intestinal parasite species. *Entamoeba histolytica/dispar*, *Giardia intestinalis*, *Ascaris lumbricoides*, and hookworm were found in 12.79%, 8.53%, 7.10%, and 1.65% of the participants, respectively. Double and triple parasite infections were found in 2.37% and 0.23% of the participants, respectively. A habit of eating unwashed vegetables (Adjusted odds ratio (AOR) = 9.98, 95% confidence interval (CI) = 2.68–37.14) and low income (AOR = 6.66, 95% CI = 1.87–23.70) were associated with increased odds of IPIs. Participants with a habit of hand washing after using the toilet had 0.05 (95% CI = 0.13–0.22) lower odds of IPIs than those who did not. In conclusion, IPIs are common among Debarq Primary Hospital patients and are associated with factors such as low income, not washing hands after using the toilet, and eating unwashed vegetation, necessitating control and prevention efforts in the study area that include health education and the provision and use of sanitary facilities.

## Introduction

Intestinal parasite infections (IPIs) continued to be one of the world's major public health problems [1, 2]. An estimated 3.5 billion people worldwide are infected with intestinal parasites [3]. Trichuriasis, amoebiasis, ascariasis, and hookworm are the most common intestinal parasitic diseases globally [4, 5]. In Sub-Saharan Africa (SSA), up to 250 million people are

**Funding:** The authors received no specific funding for this work.

**Competing interests:** The authors have declared that no competing interests exist

estimated to be infected with at least one species of intestinal parasite [6]. The prevalence of IPIs among schoolchildren was reported to be 90% in Central Sudan [7], 48.7% in Tanzania [8], 84.7% in Burkina Faso [9], and 50.0% in Rwanda [10].

Developing countries have a higher burden of IPIs than developed ones, owing to growing populations, poor environmental sanitation, inadequate toilet facilities, poverty, and climate change [5, 11]. Effective means of preventing and controlling IPIs include a combination of personal hygiene, promotion of safe methods of feces and waste disposal, improvement of general economic conditions, health education, environmental sanitation, and mass treatment of the population [12, 13].

In Ethiopia, the prevalence of IPIs has been reported to vary by region [14, 15]. For instance, infection with *Ascaris lumbricoides (A. lumbricoides)* was more common in the highlands of Ethiopia (29%), the temperate regions (35%), and the lowlands (38%) [16]. While *Trichuris trichiura* (*T..trichiura*) infection showed comparable prevalences in all altitudinal regions (13% on average), hookworm infection had the highest prevalence in the lowlands (24%), followed by temperate (15%) and highlands (7%) [14, 15]. However, the prevalence of IPIs in some areas of Ethiopia is unknown [17].

Several factors contributed to Ethiopia's high prevalence of IPIs, including low living standards, low socioeconomic status, poor personal and environmental sanitation, and unsafe human waste disposal systems [18]. It has also been reported that Ethiopia has one of the world's lowest-quality drinking water supplies and latrine coverage, which is likely to contribute to the country's high prevalence of IPIs [19]. However, in many parts of Ethiopia, the role of the above-mentioned and other factors in increasing the prevalence of IPIs is unknown [20, 21]. However, determining the prevalence of IPIs and associated risk factors aids in the development of effective preventative and control plans [22–24]. Thus, this study was conducted to determine the prevalence of IPIs and associated risk factors among patients attending Debarq Primary Hospital in northwest Ethiopia.

## Materials and methods

### Study area and population

A health facility-based cross-sectional study was conducted from March to June 2022. The study was carried out at Debarq Primary Hospital in northwest Ethiopia. Debarq town is located 282 kilometers from Bahir Dar, the capital city of the Amhara region. It is 2850 meters above sea level and lies between 13˚08′N latitude and 37˚54′E longitude. The town has a total population of 59,920 (30,615 males and 29,305 females) [25]. The inhabitants' livelihoods are built on a sustainable mixed farming system. Except for a few public health facilities, including Debarq Primary Hospital, Debarq does not have adequate access to health care [25].

### Recruitment of study participants

Patients who visited Debarq Primary Hospital for diagnosis and treatment during the study period and met the inclusion criteria were enrolled in the study at random until the required sample size was attained. Individuals who lived in the study area for at least one month before data collection and gave written consent to participate in the study were included in the study. Individuals who had recently taken antiprotozoal or anthelminthic drugs, as well as those who were critically ill during data collection, were excluded from the study.

### Sample size determination and sampling technique

The sample size was estimated using the single proportion formula [26].

$n = Z_{\alpha/2}^2 * p*(1-p) / d^2$, $n = [(1.96)^2\ 0.5\ (1-0.5)] / (0.05)^2 = 384$. Where; d is the margin of error (0.05), p is the population proportion (50%), and $Z\alpha/2$ is the normal distribution's critical value at $\alpha/2$ (for 95% CI, $\alpha$ is 0.05 and the critical value is 1.96). With a 10% non-response rate, the sample size was estimated to be 422.

## Ethical consideration

The study was conducted after obtaining ethical clearance from the University of Gondar, College of Natural and Computational Science, research ethics committee. A written consent form was used to ask for the consent of the participants and the parent or guardian of children. Participants who tested positive for any of the intestinal parasites were referred to the hospital's medical unit for treatment.

## Questionnaire survey

A pre-tested structured questionnaire was used to collect data on risk factors for intestinal parasites and the study participants' socioeconomic characteristics. The socioeconomic characteristics collected from the participants included age, gender, residence, marital status, education status, occupation, and monthly income. Participants were asked about potential risk factors for IPIs, such as access to toilets, washing hands before eating, and eating unwashed vegetables.

## Stool sample collection

Each study participant was given a clean plastic stool collection cup with an applicator stick. The participants were told to bring approximately 5 grams of their own stool. Each stool collection cup is labeled with the date of sample collection and participant's name or number. The stool samples collected from each participant were examined within 30 minutes of collection at Debarq Primary Hospital's parasitology laboratory. The stool examination was done independently by two qualified laboratory technicians. All reagents and supplies used to collect and analyze stool samples were checked for quality.

## Stool examination

**Direct wet / iodine mount.** Following a macroscopic examination of the stool samples for consistency, color, and the presence of blood, mucus, and adult intestinal helminths, a direct wet mount was prepared and examined as described elsewhere [27]. Briefly, approximately 2 grams of stool were emulsified with normal saline (0.85% sodium chloride (NaCl) solution), and then a drop of the emulsified sample was placed on a clean microscopic glass slide. An iodine mount was made on the other side of the slide. Coverslips were placed at a 45˚ angle to cover both wet mounts. Finally, the samples were examined under a light microscope with 10x and 40x objective lenses for larvae, cysts, eggs and trophozoites of motile intestinal parasites.

**Formal-ether concentration.** In addition to the direct wet mount, supplementary procedures such as concentration techniques are used to recover the eggs of parasites such as schistosoma, which are found in low concentration in stool and cannot be detected by wet mount [27]. A portion of the stool samples collected from the participants was examined using the formal-ether concentration technique as described elsewhere [28]. Briefly, about 1–2 grams of stool samples were placed in a clean 15 ml conical tube containing 7 ml of 10% formalin using a wooden applicator stick. After mixing the sample with the applicant, the suspension was filtered through a sieve (cotton gauze) into a beaker, and the filtrate was placed in the same tube. After adding 3 mL of diethyl ether to the mixture and hand-shaking it, the contents were

centrifuged at 2000 rpm for 3 minutes. The sediments were used to prepare an iodine stain. Finally, the entire area under the coverslip was examined with 10x and 40x objective lenses. Specimens were considered positive if helminth eggs, larvae, cysts and/or protozoan trophozoites were found using either wet/iodine mount or formal ether concentration techniques.

## Data analysis

The data was entered and analyzed using the statistical package for social sciences (SPSS) software version 25 (Armonk, NY: IBM Corp.). Crude odds ratio (COR) and adjusted odds ratio (AOR) were calculated using logistic regression to examine the association between socioeconomic characteristics, risk factors for intestinal parasites, and IPIs. Variables with $p < 0.25$ in univariable logistic regression were included in the multivariable logistic regression. The references for logistic regression were chosen based on the expected risk for IPIs. P-values $\leq 0.05$ were considered statistically significant.

## Results

### Characteristics of the study participants

Table 1 shows the socio-economic characteristics of the study participants. A total of 422 individuals (212 males and 210 females) participated in the study. The majority of the participants (21.80%) were aged 40–49. While 57.35% of participants lived in urban areas, 42.65% lived in rural areas. The majority of participants (41.94%) were married, 43.13% were illiterate, and 43.84% had a household size of 4 to 6, and 31.04% were government employees.

### Prevalence of intestinal parasitic infections

The prevalence of single, double, and triple parasite infections among the study participants is shown in Table 2. Seven intestinal parasites (two protozoans and five helminths) were identified in the study participants. In 33.64% of the participants, single, double, or triple parasite infections were identified. Single and double parasite infections were identified in 66 (15.64%) and 6 (1.42%) of the male participants, respectively. In females, 65 (15.40%) had a single parasite infection, whereas 5 (0.95%) had a double infection.

The most commonly found parasite identified in the study participants was *E. histolytica/dispar* 54 (12.79%), followed by *G. intestinalis* 36 (8.53%), *A. lumbricoides* 30 (7.10%), hookworm 7(1.65%), *S. mansoni* 3 (0.71%), and *H. nana* 1(0.23%). *E. histolytica/dispar* and *G. intestinalis* caused the highest proportion 4(0.94%) of double infections. *E. histolytica/dispar*, *G. intestinalis*, and *T. trichiura* were found in only 0.23% of female participants.

### Logistic regression analysis for potential risk factors of intestinal parasitic infections

Table 3 shows a logistic regression analysis of factors affecting IPIs. Participants under the age of 10 were significantly less likely to be infected with intestinal parasites than the other age groups. There were higher odds of IPIs in those with a monthly income of less than 1000 ETB per month compared to those earning more than 3000 ETB per month (AOR = 6.66, CI = 1.87–23.70). Participants who wash their hands after using toilets were 0.05 (0.13–0.22) less likely to have IPIs than those who did not. IPIs were associated with a habit of eating unwashed vegetables (AOR = 9.98, CI = 2.68–37.14). Participants with a family size of 4 to 6 were 25.74 (3.10–213.44) times more likely to have IPIs than those with a family size of $\leq 3$.

**Table 1. General characteristics of the study participants (N = 422).**

| Variables | Categories | Number | Percentage |
|---|---|---|---|
| Sex | Male | 212 | 50.20 |
| | Female | 210 | 49.80 |
| Age group (years) | < 10 | 70 | 16.59 |
| | 10–19 | 41 | 9.72 |
| | 20–29 | 38 | 9.00 |
| | 30–39 | 24 | 5.69 |
| | 40–49 | 92 | 21.80 |
| | 50–59 | 42 | 9.95 |
| | 60–69 | 88 | 20.8 |
| | ≥ 70 | 27 | 6.40 |
| Residence | Urban | 242 | 57.35 |
| | Rural | 180 | 42.65 |
| Marital status | Married | 177 | 41.94 |
| | Single | 144 | 34.12 |
| | Divorced | 87 | 20.62 |
| | Widowed | 14 | 17.54 |
| Education level | Illiterate | 178 | 42.18 |
| | Primary or secondary school | 192 | 45.50 |
| | Diploma and above | 52 | 12.32 |
| Occupational status | Merchants | 50 | 11.85 |
| | Government employers | 131 | 31.04 |
| | Housewives | 66 | 15.64 |
| | Students | 91 | 21.56 |
| | Farmers | 84 | 19.91 |
| Family size | 4–6 | 185 | 43.84 |
| | >6 | 130 | 30.81 |
| | ≤3 | 107 | 25.36 |
| Monthly Income (ETB) | > 3000 | 21 | 4.98 |
| | 1000–3000 | 179 | 42.42 |
| | < 1000 | 222 | 52.61 |

ETB = Ethiopian birr.

## Discussion

Determining the magnitude of IPIs and identifying associated risk factors in a certain population is important for the development, implementation, and evaluation of preventive and control measures [12, 13]. This study assessed the prevalence of IPIs and associated risk factors among patients attending Debarq Primary Hospital in northwest Ethiopia. The study found a high prevalence of IPIs among patients at Debarq Primary Hospital. Low income, eating unwashed vegetables, not washing hands after using the toilet, and family size were also found to be associated with increased odds of IPIs.

The overall prevalence of IPIs in this study (33.64%) is lower than a report from Aksum, north Ethiopia (44.6%) [29], but higher than that found in southern Ethiopia (26.2%) [30]. A higher prevalence of intestinal parasitic infections has also been reported from other African countries such as Nigeria (63.5%) [31], Sudan (62.5%) [32], and Kenya (53.8%) [33]. Poor environmental sanitation, overpopulation, and low altitude have all been associated with an increase in the prevalence of IPIs [34]. Moreover, the prevalence of IPIs has been associated

**Table 2. Intestinal parasite infections among study participants by gender (N = 422).**

| Types of intestinal parasites | Sex | | |
|---|---|---|---|
| | **Male no. (%)** | **Female no. (%)** | **Total no. (%)** |
| *E.histolytica/dispar* | 22(5.21) | 32(7.58) | 54(12.79) |
| *G.intestinalis* | 23(5.45) | 13(3.08) | 36(8.53) |
| *A.lumbricoides* | 11(2.60) | 19(4.50) | 30(7.10) |
| Hookworm | 6(1.42) | 1(0.23) | 7(1.65) |
| *Heymenolipis nana* (*H.nana*) | 1(0.23) | 0(0.00) | 1(0.23) |
| *Schistosoma mansoni* (*S.mansoni*) | 3(0.71) | 0(0.00) | 3(0.71) |
| Total | 66(15.64) | 65(15.40) | 131(31.04) |
| Double infection (n = 10) | | | |
| *E. histolytica/dispar* & *G.intestinalis* | 3(0.71) | 1(0.23) | 4(0.95) |
| *E. histolytica/dispar* & *A. lumbricoides* | 0(0.00) | 1(0.23) | 1(0.23) |
| *E. histolytica/dispar* & hookworm | 1(0.23) | 0(0.00) | 1(0.23) |
| *E. histolytica/dispar* & *S.mansoni* | 1(0.23) | 0(0.00) | 1(0.23) |
| *E. histolytica/dispar* & *T. trichiura* | 0 | 1(0.23) | 1(0.23) |
| *G. intestinalis* & hookworm | 0(0.23) | 1(0.23) | 1(0.23) |
| *A. lumbricoides* & hookworm | 1(0.23) | 0(0.00) | 1(0.23) |
| Total | 6(1.42) | 4(0.95) | 10(2.37) |
| Triple infection (n = 1) | | | |
| *E.histolytica/dispar* &*G.intestinalis* & *T.trichiura* | 0(0.00) | 1(0.23) | 1(0.23) |
| Total | 0(0.00) | 1(0.23) | 1(0.23) |

with differences in climatic conditions, the cultural practices of the study participants, and previous control measures [21]. The differences in the prevalence of IPIs between this study and other studies in Ethiopia and other African countries may be explained by variations in the aforementioned factors.

The predominant intestinal parasite identified in the present study was *E. histolytica/dispar* (12.79%), followed by *G. intestinalis* (8.53%). These parasites also formed the most concurrently appearing mixed infections in this study, which is consistent with findings in northwest and southern Ethiopia [5, 35]. The high prevalence of *E. histolytica/dispar* and *G. intestinalis* found in this study is consistent with a World Health Organization (WHO) report that identified these two parasites as prevalent causes of intestinal infections throughout Ethiopia [12]. *E. histolytica/dispar* and *G. intestinalis* have also been shown to be common causes of intestinal infection in various countries, including South Africa [36], Indonesia [37], and India [38]. The high prevalence of these two protozoan parasites might be explained by the fact that they have a feco-oral transmission route and higher reproduction and persistence capacities in the environment and the host [39, 40]. The cysts of *E. histolytica/dispar* are highly resistant to normal concentrations of chlorine and can persist in water at 0˚C for several months [39, 40].

In Ethiopia, inadequate sanitary facilities such as toilets and latrines pose a significant concern [41, 42]. The percentage of improved toilet use (not shared) is merely 6% [43]. This has been reported to encourage open defecation, contaminating the country's water sources [41, 42, 44]. It has been reported that 8.7% of urban and 37.5% of rural residents in Ethiopia engaged in open defecation, and 82.5% of urban and 97.5% of rural residents lacked access to adequate sanitation [45]. Such contamination encourages the spread of intestinal parasites [6]. This could explain the high prevalence of IPIs in the study area; hence, recognizing and addressing sanitary facility concerns in the study area is crucial to reducing the incidence of IPIs.

**Table 3. Multivariable logistic regression analysis for potential determinants of intestinal parasitic infections among the participants (N = 422).**

| Risk factors | Categories | N (%) | IPIs | | COR (95%CI) | P- value | AOR (95%CI) | P-value |
|---|---|---|---|---|---|---|---|---|
| | | | Positive (%) | Negative (%) | | | | |
| Sex | Male | 212 (50.24) | 72 (17.06) | 140 (33.18) | 1.02 (0.68–1.54) | 0.89 | NA | NA |
| | Female | 210 (49.76) | 70 (16.59) | 140 (33.18) | 1 | | | |
| Family size | >6 | 130 (30.81) | 36(8.53) | 94(22.27) | 1.97(1.15–3.39) | 0.01* | 25.74(3.10–213.44) | 0.003* |
| | ≤3 | 107 (25.36) | 46(10.90) | 61(14.45) | 1.25(0.77–2.05) | 0.37 | 1.17(0.25–5.38) | 0.84 |
| | 4–6 | 185 (43.84) | 60(14.22) | 125(29.62) | 1 | | 1 | |
| Residence | Urban | 242 (57.35) | 81 (19.19) | 161(38.15) | 1 | | | |
| | Rural | 180 (42.65) | 61 (14.45) | 119(28.20) | 1.02 (0.68–1.53) | 0.93 | NA | NA |
| Age group | < 10 | 70 (16.59) | 31 (7.35) | 39(9.24) | 1 | | 1 | |
| | 10–19 | 41 (9.72) | 8(1.90) | 33(7.82) | 0.44 (0.17–1.18) | 0.10 | 25.03(1.11–566.280) | 0.04* |
| | 20–29 | 38(9.00) | 18 (4.27) | 20 (4.74) | 0.31 (0.12–0.75) | 0.01* | 21.29(0.95–476.22) | 0.04* |
| | 30–39 | 24(5.69) | 14(3.32) | 10(2.37) | 0.63 (0.28–1.40) | 0.25 | 54.25(2.43–1210.00) | 0,01* |
| | 40–49 | 92(21.80) | 30(7.11) | 62(14.69) | 1.13 (0.51–2.50) | 0.76 | 84.40(5.09–1397.68) | 0.002* |
| | 50–59 | 42(9.95) | 14(3.32) | 28(6.64) | 1.76 (0.69–4.50) | 0.24 | 64.42(4.18–10.25) | 0.003* |
| | 60–69 | 88(20.88) | 20(4.74) | 68(16.11) | 0.61 (0.32–1.16) | 0.13 | 38.9(2.7–553.58) | 0.007* |
| | ≥ 70 | 27(6.40) | 7(1.66) | 20(4.74) | 0.37 (0.19–0.74) | 0.005* | 6.92(1.31–36.66) | 0.02* |
| Education status | Illiterate | 178 (42.18) | 61 (14.45) | 125 (29.62) | 0.91(0.46–1.76) | 0.78 | NA | NA |
| | Primary or secondary school | 192 (45.50) | 70 (16.59) | 122 (28.91) | 1.18(1.77–1.80) | 0.77 | NA | NA |
| | Diploma and above | 52 (12.32) | 16 (3.79) | 36 (8.53) | 1 | | | |
| Occupation | Merchants | 50(11.85) | 10(2.37) | 40(9.48) | 0.67(0.36–1.23) | 0.19 | 0.21(0.38–1.15) | 0.07 |
| | Government employers | 131 (31.04) | 52(12.32) | 79(18.72) | 0.31(0.14–0.69) | 0.004* | 0.14(0.03–0.74) | 0.21 |
| | Housewives | 66(15.64) | 10(2.37) | 56(13.27) | 0.82(0.48–1.41) | 0.47 | 0.56(0.11–2.79) | 0.48 |
| | Students | 91(21.56) | 41(9.72) | 50(11.85) | 0.22(0.10–0.49) | <0.01* | 0.06 (0.10–0.29) | 0.001* |
| | Farmers | 84(19.91) | 29(6.87) | 55(13.03) | 1 | | 1 | |
| Marital status | Married | 177 (41.94) | 74(17.54) | 103(24.41) | 1 | | | |
| | Single | 144 (34.12) | 41(9.72) | 103(24.41) | 1.29 (0.42–4.02) | 0.66 | NA | NA |
| | Divorced | 87(20.62) | 65(15.40) | 22(5.21) | 0.72 (0.23–2.27) | 0.57 | NA | NA |
| | Widowed | 14(17.54) | 9(2.13) | 5(1.18) | 0.18 (0.18–2.01) | 0.42 | NA | NA |

(*Continued*)

**Table 3.** (*Continued*)

| Risk factors | Categories | N (%) | IPIs | | COR (95%CI) | P- value | AOR (95%CI) | P-value |
|---|---|---|---|---|---|---|---|---|
| | | | Positive (%) | Negative (%) | | | | |
| Monthly income (ETB) | < 1000 | 222 (52.61) | 70(16.59) | 152(36.02) | 4.34(1.68–11.23) | 0.002* | 6.66(1.87–23.70) | 0.003* |
| | 1000–3000 | 179 (42.42) | 58(13.74) | 121(28.67) | 1.04 (0.68–1.59) | 0.85 | 1.95(0.84–4.55) | 0.12 |
| | > 3000 | 21 (4.98) | 14(3.32) | 7(1.66) | 1 | | 1 | |
| A habit of hand washing after using the toilet | No | 237 (56.20) | 78(18.50) | 159(37.70) | 1 | | 1 | |
| | Yes | 185 (43.80) | 64(28.70) | 121(15.20) | 0.93(0.62–1.39) | 0.72 | 0.05(0.13–0.22) | < 0.01* |
| A habit of eating unwashed vegetable | Yes | 197 (46.70) | 74(29.10) | 74(17.50) | 1.39 (0.93–2.08) | 0.11 | 9.98(2.68–37.14) | 0.001* |
| | No | 225 (53.30) | 68(16.10) | 157(37.20) | 1 | | 1 | |
| Presence of latrine at home | Yes | 187 (44.30) | 56(13.30) | 131(31.00) | | | 1 | |
| | No | 235 (55.70) | 86(35.30) | 86(20.40) | 1.35 (0.89–2.04) | 0.15 | 1.32 (0.72–2.44) | 0.37 |
| Hand washing before food | No | 406 (96.20) | 131 (31.0) | 275(65.20) | 1 | | 1 | |
| | Yes | 16(3.80) | 11(2.60) | 5(1.20) | 0.26 (0.94–0.72) | 0.01* | 0.76(0.23–2.58) | 0.67 |

1, reference value

*, statistically significant; COR, crude odds ratio; N, total number of cases; ETB, Ethiopian Birr; IPIs, intestinal parasite infections; 95% CI, 95% confidence interval; NA, not applicable.

Many individuals in Ethiopia lack adequate knowledge of good hygiene practices such as handwashing, which can lead to the spread of intestinal parasites [44]. In 2013, the Ethiopian government started the WASH National Program, with some of its goals being reducing open defecation and enhancing public sanitation, such as hand washing and improving sanitary facilities [44, 46]. The program has had considerable results, including a 17% reduction in open defections and an improvement in hygienic practices in schools and households[41, 44, 46]. However, there are several areas in the country where such a program has received little attention [41, 44, 46]. Thus, implementing this program in areas where it has received little attention is likely to reduce the prevalence of IPIs.

In this study, a low income (less than ETB 1,000) was associated with increased odds of IPIs, which is consistent with the findings of other similar studies [47–50]. Income has also been associated with an increased risk of IPIs in other regions of Ethiopia [17, 22, 51, 52], and in another country [53]. People with higher incomes have higher access to sanitation supplies such as soap, toilets, and other facilities, which reduces their chances of contracting intestinal parasites [54–56]. Low income has also been linked to inadequate sewage network coverage and bad environmental conditions [57], such as living in overcrowded houses with limited internal space and in shantytowns [58]. The aforementioned factors could explain why low-income participants in our study have higher odds of IPIs.

The finding in this study that IPIs are associated with not washing after using the bathroom is consistent with findings in other parts of Ethiopia and elsewhere [59–62]. This could be because poor personal hygiene has been linked to the spread of intestinal parasites [63]. More-over, this study found that eating unwashed vegetables is associated with an increased risk of

IPIs. This is in line with studies conducted in other regions of Ethiopia [64, 65]. Fruits and vegetables can be contaminated with the infective stages of intestinal parasites, and eating them without washing increases the risk of infection [66–68].

This study also found that participants with a family size of more than six had higher odds of IPIs than those with family sizes of three or less. This is consistent with previous studies in Ethiopia [69, 70]. A study in another African country also reported an increased risk of IPIs in individuals with large family sizes [71]. This study's findings could be explained by the fact that an overcrowded household situation increases the chance of parasite transmission [70] Moreover, parents with large families may not have enough time to care for their children, increasing parasite transmission. Large family sizes have also been linked to decreased access to basic sanitary facilities due to resource sharing, which increases the risk of IPIs [69].

The findings of this study should be considered alongside its limitations. First, because this is a health-care facility-based study, its findings may not represent the prevalence of IPIs and associated risk factors in the general population. Second, this study did not assess all of the potential risk factors for IPIs, emphasizing the need for further similar studies that take into account all of the potential risk factors for IPIs. Moreover, the present study relied only on smear microscopy in the identification of the parasites, which has lower sensitivity than molecular methods to differentiate species such as *E. histolytica/dispar* and *E. dispar*.

## Conclusions

The prevalence of IPIs in patients attending Debarq Primary Hospital was found to be relatively high, with protozoan parasites (*E. histolytica/dispar and G. intestinalis*) being predominantly identified. IPIs were found to be associated with age, low income, being a student, not washing hands after using toilets, and eating unwashed vegetation. It is, therefore, important to implement control and preventative measures in the study region that include health promotion as well as the availability and use of sanitary facilities. More community based research with better diagnostic techniques on IPIs is necessary to comprehend the epidemiology of the infections and to design and implement appropriate parasite infection prevention measures in the study area in particular.

## Supporting information

**S1 Checklist. STROBE statement—checklist of items that should be included in reports of observational studies.**
(DOCX)

## Acknowledgments

We would like to thank the data collectors, study participants, and staff at Debarq Primary Hospital for their cooperation.

## Author Contributions

**Conceptualization:** Amir Alelign.

**Data curation:** Amir Alelign, Zinaye Tekeste.

**Formal analysis:** Amir Alelign, Nigus Mulualem, Zinaye Tekeste.

**Investigation:** Amir Alelign, Nigus Mulualem.

**Methodology:** Amir Alelign.

**Supervision:** Amir Alelign.

**Writing – original draft:** Amir Alelign, Nigus Mulualem.

**Writing – review & editing:** Amir Alelign, Zinaye Tekeste.

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
