## [Decision Letter · Decision Letter 0]

24 Jul 2023

PONE-D-23-15672Prevalence of human intestinal parasitic infections and associated risk factors in Debarq, northwest EthiopiaPLOS ONE

Dear Dr. Alelign,

Thank you for submitting your manuscript to PLOS ONE. After careful consideration, we feel that it has merit but does not fully meet PLOS ONE’s publication criteria as it currently stands. Therefore, we invite you to submit a revised version of the manuscript that addresses the points raised during the review process.

Please find the comments provided by the editor and reviewers below this email.

We look forward to receiving your revised manuscript.

Kind regards,

Hesham

Hesham M. Al-Mekhlafi, PhD

Academic Editor

PLOS ONE

Additional Editor Comments:

Academic editor's comments:

Dear Amir,‎

The reviewers have provided some comments that should be adequately addressed addressed. Moreover, I've corrected few errors and added some comments on the manuscript, using comments ‎and ‎track changes in the attached annotated file. ‎The file can be downloaded from your author center. Please, accept changes, where applicable, and address the comments in your revised manuscript, where made. For instance (refer to annotated manuscript file attached with this email):

1- Analysis should be revised and all related data should be provided in Tables 3 and 4.

2- Why occupation, family size and residence were not included in multivariate analysis? They showed P values <0.25.

3- Additional limitations of the study should be acknowledge; e.g. not using Kato-Katz to estimate intensity of infections; study design (hospital-based); among symptomatic patients; generalizability of findings; etc.

4- Other concerns that should be addressed; e.g. clarification about covering some variables (e.g. clean drinking water lines 222-230) and rationale for age grouping; etc.

Reviewers' comments:

Reviewer's Responses to Questions

**Comments to the Author**

1. Is the manuscript technically sound, and do the data support the conclusions?

Reviewer #1: Yes

Reviewer #2: Yes

2. Has the statistical analysis been performed appropriately and rigorously? 

Reviewer #1: I Don't Know

Reviewer #2: Yes

3. Have the authors made all data underlying the findings in their manuscript fully available?

Reviewer #1: Yes

Reviewer #2: Yes

4. Is the manuscript presented in an intelligible fashion and written in standard English?

Reviewer #1: Yes

Reviewer #2: No

5. Review Comments to the Author

Reviewer #1: The manuscript is interesting, well designed, and their results has important for public health. I gave some minor comments:

1) If the some results demonstrate as charts or graphs, the results could have more visibility, then you can your table as supplementary materials.

Reviewer #2: Prevalence of human intestinal parasitic infections and associated risk factors in Debarq, northwest Ethiopia

1. Abstract- Line 30; how where the study participants recruited

2. Introduction

3. In paragraph one please add the African burden of IIPs

4. Line 55- was should be replaced with have

5. Line 57- T.trichiura is appearing for the first time and should be written in full.

6. Line 61-63 is a repletion of line 50-52 and should be deleted

7. Line 68-69 that statement is not true

8. The introduction can be improved, it is information about the local context what about elsewhere? What are the current WHO recommendations about prevention and control of the parasites?

9. Methods

10. Sub -heading inclusion criteria is not needed in manuscript writing- can sampling procedure/recruitment of study participants

11. Sub heading 100- can be replace with questionnaire survey

12. Line 101; how where the study participants recruited?

13. Results

14. Table three can be replaced with table 4

15. Discussion

16. Line 212- what about other countries with similar problems; what is their prevalence level?

17. The discussion is very scanty- most of the associated risk factors have not been discussed and a reason given why they were associated and what others have found with similar studies.

18. The discussion needs a lot of revision- all the associated risk factors needs to be discussed.

19. Addition of qualitative data would have given more insight as to why this IIPs are still a problem.

6. PLOS authors have the option to publish the peer review history of their article (what does this mean?). If published, this will include your full peer review and any attached files.

Reviewer #1: No

Reviewer #2: No

---

## [Author Response · Author response to Decision Letter 0]

21 Aug 2023

Prevalence of human intestinal parasitic infections and associated risk factors in Debarq, northwest Ethiopia

Reviewer #2

Dear Reviewer,

Thank you for your suggestions and comments. Please find a point-by-point response to your comments and suggestions below.

1. Abstract- Line 30; how where the study participants recruited

• As suggested, the recruitment process is described in the abstract and methods sections.

2. Introduction

3. In paragraph one please add the African burden of IIPs

• The African burden of IPIs is described as suggested.

4. Line 55- was should be replaced with have

• Corrected as suggested. 

5. Line 57- T.trichiura is appearing for the first time and should be written in full.

• Corrected as suggested. 

6. Line 61-63 is a repletion of line 50-52 and should be deleted

• Lines 50-52 are about factors for IPIs in developing nations, and lines 61-63 are about factors for IPIs in Ethiopia. Because Ethiopia is a developing country, it shares factors with other developing countries.

7. Line 68-69 that statement is not true

• Corrected as suggested.

8. The introduction can be improved, it is information about the local context what about elsewhere? What are the current WHO recommendations about prevention and control of the parasites?

• Information on IPIs from elsewhere is included, as well as information on parasite prevention and control.

9. Methods

10. Sub -heading inclusion criteria is not needed in manuscript writing- can sampling procedure/recruitment of study participants

• The subheading is replaced with recruitment of study participants

11. Sub heading 100- can be replace with questionnaire survey

• The sub-heading is replaces with questionnaire survey as suggested.

12. Line 101; how where the study participants recruited?

• The recruitment process is described as suggested.

13. Results

14. Table three can be replaced with table 4

• Tables 3 and 4 were merged.

15. Discussion

16. Line 212- what about other countries with similar problems; what is their prevalence level?

• The prevalence of IPIs in other countries is described as suggested.

17. The discussion is very scanty- most of the associated risk factors have not been discussed and a reason given why they were associated and what others have found with similar studies.

• All of the associated factors are explored in light of your suggestions. The entire portion of discussion is revised.

18. The discussion needs a lot of revision- all the associated risk factors needs to be discussed.

• All of the associated factors are explored in light of your suggestions. The entire portion of discussion is revised.

19. Addition of qualitative data would have given more insight as to why this IIPs are still a problem.

• Because the study is quantitative and lacks qualitative data, we are unable to include the data you proposed in the revised version of the manuscript.

Reviewer #1

Dear Reviewer,

Thank you for your suggestions and comments. Please find a point-by-point response to your comments and suggestions below.

1) If the some results demonstrate as charts or graphs, the results could have more visibility, then you can your table as supplementary materials.

• Because of the nature of the data, including its being with many variables, we assumed that placing it in a table rather than charts and graphs would describe it more clearly and concisely.

---

## [Decision Letter · Decision Letter 1]

8 Jan 2024

PONE-D-23-15672R1Prevalence of intestinal parasitic infections and associated risk factors among patients attending Debarq Primary Hospital, northwest EthiopiaPLOS ONE

Dear Dr. Alelign,

Thank you for submitting your manuscript to PLOS ONE. After careful consideration, we feel that it has merit but does not fully meet PLOS ONE’s publication criteria as it currently stands. Therefore, we invite you to submit a revised version of the manuscript that addresses the points raised during the review process.

We look forward to receiving your revised manuscript.

Kind regards,

Gideon Zulu, MD, MPH

Academic Editor

PLOS ONE

Journal Requirements:

Reviewers' comments:

Reviewer's Responses to Questions

**Comments to the Author**

1. If the authors have adequately addressed your comments raised in a previous round of review and you feel that this manuscript is now acceptable for publication, you may indicate that here to bypass the “Comments to the Author” section, enter your conflict of interest statement in the “Confidential to Editor” section, and submit your "Accept" recommendation.

Reviewer #1: All comments have been addressed

Reviewer #3: (No Response)

2. Is the manuscript technically sound, and do the data support the conclusions?

Reviewer #1: Yes

Reviewer #3: Yes

3. Has the statistical analysis been performed appropriately and rigorously? 

Reviewer #1: Yes

Reviewer #3: Yes

4. Have the authors made all data underlying the findings in their manuscript fully available?

Reviewer #1: Yes

Reviewer #3: No

5. Is the manuscript presented in an intelligible fashion and written in standard English?

Reviewer #1: Yes

Reviewer #3: Yes

6. Review Comments to the Author

Reviewer #1: Thank you for your revision.

The authors revised the manuscript sufficiently. In my opinion, the manuscript is merit for publication.

Reviewer #3: No raw data has been provided, not even the tools used to collect data. Thus I would have liked to see the questionnaires. Infact, this study does not show that it was ethically approved.

7. PLOS authors have the option to publish the peer review history of their article (what does this mean?). If published, this will include your full peer review and any attached files.

Reviewer #1: No

Reviewer #3: No

---

## [Author Response · Author response to Decision Letter 1]

17 Jan 2024

Response to the Academic Editor and Reviewers’ Comments

Prevalence of human intestinal parasitic infections and associated risk factors in Debarq, northwest Ethiopia

Academic Editor

Dear Academic editor of the journal,

Thank you very much for your suggestions and concerns on our manuscript to fit with the journal requirements. As per your suggestion on the reference list of the manuscript, we have gone through it and made the following changes:

• Every reference on the list have been checked and corrected to meet with the journal’s standard requirements. The changes are highlighted and indicated in the track change version of the revised manuscript. 

• An additional reference (reference number 27) has been incorporated to the reference list and the citation numbers in the main text of the manuscript are corrected accordingly. 

Reviewer #3

Dear Reviewer,

Thank you for your suggestions and comments. Please find a point-by-point response to your comments and suggestions below.

Introduction: 

1. Line 57. Put space between T and full stop to read as T. trichiura

• Corrected as suggested and indicated in line 60

Materials and methods:

Study area and population:

2. Line 80: Leave a space between longitude.The

• Corrected by leaving a space as suggested, indicated in line 79

Inclusion and exclusion criteria: 

3. Each of these criteria to have its own paragraph. 

• As we have been convinced by the suggestion from the previous reviewer, the sub-heading ‘Inclusion/Exclusion criteria is replaced with a more appropriate sub –heading for a manuscript which is ‘Recruitment of study participants’. The criteria for inclusion/exclusion are also included in this sub-heading.

Stool examination:

Direct wet / iodine mount:

4. Line 122: replace “was” with “were” …………, in approximately 2 grams of stool was emulsified……………

• ‘’was’’ is replaced with ‘’were’’ as suggested, indicated in line 124

5. Line 127: delete comma after eggs in ………………… cysts, eggs, and trophozoites of motile intestinal parasites.

• Corrected as suggested by the reviewer, indicated in line 129

Formal-ether concentration:

6. Line 137: Delete comma after eggs in ………………if helminth eggs, larvae, cysts, and/or protozoan trophozoites

• Comma after ‘cyst’ is deleted, but we found the comma placed after ‘eggs’ is appropriate as indicated in line 139.

7. Question: Why did the authors use two methods of stool analysis without showing the results for each of the methods used? I do not see the need of including analysis whose results you do not use. Unless, they present the results for each of the methods used, they need to only pick one method.

• In our study, we have used ‘wet mount’ as an obligatory technique for the initial phase of intestinal parasites microscopic examination. In case of samples negative for intestinal parasites in the wet mount, we have used the formol-ether concentration technique. Particularly, to recover the eggs of parasites such as Schistosomes, which are naturally found in low concentration in stool and cannot be detected by wet mount, the above supplementary method was useful. Hence, we have used both techniques in our study. The need for the two methods in our study is justified and highlighted in the track-change version of the revised manuscript, indicated from lines 131 to 133. 

Moreover, as the study aimed to determine the magnitude of the intestinal parasites infection in the study area, not to compare the performance of the two methods, we have assumed that presenting the overall parasite distribution would be more appropriate. 

Results:

Characteristics of the study participants:

8. Lines 150 and 151: Add “d” on the two words of “live”) ……………While 57.35% of participants live in urban areas, 42.65% live in rural areas. The results have to be stated in a reported speech.

• Corrected as suggested by the reviewer, indicated in lines 156 and 157.

Prevalence of intestinal parasitic infections:

9. Line 162: Add “mostly” between “and “and “in”……in participants aged 25 and older and in men (17.06%).

• The statement is replaced with a more appropriate data presentation, as highlighted from lines 164-167. 

Discussion:

10. Lines 218 to 219: Capitalise each word of the…………… world health organization (WHO) to read as ………….World Health Organization”…………..

• Corrected as suggested and that portion of the discussion is revised as indicated from lines 215-219.

11. Line 221: Replace “norm” with “normal” ………………resistant to the norm concentration of chlorine in drinking water

• Corrected in light of your suggestion as highlighted in lines 222 & 223. 

12. Line 229: …………. area, such issues must be identified and addressed. Please note that this study has already identified them and I guess what is needed now is to address them. Thus delete: “must be identified and”

• Corrected as suggested and highlighted from lines 229-231.

13. Line 233: Add “were” between residents and engaged in ……………….. and 37.5% of rural residents engaged in open defecation,

• In light of your suggestion, the paragraph is modified in a more appropriate manner

14. Line 237: See previous comment made on Line 229. Please delete “identify and” in ………………. it is essential to identify and address sanitary facility concerns in…………..

• As per your suggestion, the statements are rewritten in a more appropriate way

15. Line 248: Replace “are” with “is” in …………….. launching similar programs are likely to reduce the prevalence of intestinal………… because this sentence is talking about the implementation of the WASH programme.

• The statement is rewritten to incorporate more relevant concepts

16. Line 259: Add “be” between to and associate in …………..has also been found to associate with an increased risk of intestinal………….. Further, replace “associate” with “associated”

• Corrected as suggested with modification to the statement

17. Line 273: Replace “are” with “being” in ……………….with protozoan parasites (E. histolytica and G. lamblia) are predominantly…………

• Corrected as suggested and highlighted in lines 273 & 274 

18. Line 274: Delete “a” on “abe” in ……………… of intestinal parasites was observed to abe associated…………………….

• Corrected as per your suggestion and highlighted in line 275 of the track change version of the revised manuscript. 

Reviewer #1

Dear Reviewer,

Thank you for your suggestions and comments. Please find a point-by-point response to your comments and suggestions below.

1. Stool sample collection- Line 113-t0 155 can be rephrased. Also, in line 144, use past tense instead of present tense.

• We understand that the comment was for lines 133 to 115 (not 155). Hence, we have tried to rephrase the statement as per your suggestion and highlighted, as indicated from lines 113-116 of the track version of the revised manuscript.

• Unfortunately, we could not get present tens in line 144 as commented by the reviewer; hence, we were not able to correct it as suggested. 

2. The conclusion can be improved by a more detailed recommendation. Health education can be changed to health promotion. 

• In agreement with your suggestion, we tried to improve the conclusion with additional recommendation points as highlighted and indicated from lines 279-282. 

• As suggested, health ‘education’ is replaced with health ‘promotion’, indicated/highlighted in line 278 of the track version of the revised manuscript. 

Thank you All,

The corresponding author of the manuscript

---

## [Decision Letter · Decision Letter 2]

31 Jan 2024

Prevalence of intestinal parasitic infections and associated risk factors among patients attending Debarq Primary Hospital, northwest Ethiopia

PONE-D-23-15672R2

Dear Dr. Alelign,

We’re pleased to inform you that your manuscript has been judged scientifically suitable for publication and will be formally accepted for publication once it meets all outstanding technical requirements.

Kind regards,

Gideon Zulu, MD, MPH

Academic Editor

PLOS ONE

Additional Editor Comments (optional):

Reviewers' comments:

Reviewer's Responses to Questions

**Comments to the Author**

1. If the authors have adequately addressed your comments raised in a previous round of review and you feel that this manuscript is now acceptable for publication, you may indicate that here to bypass the “Comments to the Author” section, enter your conflict of interest statement in the “Confidential to Editor” section, and submit your "Accept" recommendation.

Reviewer #3: All comments have been addressed

2. Is the manuscript technically sound, and do the data support the conclusions?

Reviewer #3: Yes

3. Has the statistical analysis been performed appropriately and rigorously? 

Reviewer #3: Yes

4. Have the authors made all data underlying the findings in their manuscript fully available?

Reviewer #3: Yes

5. Is the manuscript presented in an intelligible fashion and written in standard English?

Reviewer #3: Yes

6. Review Comments to the Author

Reviewer #3: In as much as you attended to most of the concerns raised, It would have been better if you included the reference number for the ethical approval made by your university.

7. PLOS authors have the option to publish the peer review history of their article (what does this mean?). If published, this will include your full peer review and any attached files.

Reviewer #3: No

---

## [Editor Report · Acceptance letter]

26 Feb 2024

PONE-D-23-15672R2 

PLOS ONE

Dear Dr. Alelign, 

I'm pleased to inform you that your manuscript has been deemed suitable for publication in PLOS ONE. Congratulations! Your manuscript is now being handed over to our production team.

Kind regards, 

on behalf of

Dr Gideon Zulu 

Academic Editor

PLOS ONE